# Cold Case: the Lost MNIST Digits

**Chhavi Yadav**
New York University
New York, NY
chhavi@nyu.edu

**Léon Bottou**
Facebook AI Research
and New York University
New York, NY
leon@bottou.org

## Abstract

Although the popular MNIST dataset [LeCun et al., 1994] is derived from the NIST database [Grother and Hanaoka, 1995], the precise processing steps for this derivation have been lost to time. We propose a reconstruction that is accurate enough to serve as a replacement for the MNIST dataset, with insignificant changes in accuracy. We trace each MNIST digit to its NIST source and its rich metadata such as writer identifier, partition identifier, etc. We also reconstruct the complete MNIST test set with 60,000 samples instead of the usual 10,000. Since the balance 50,000 were never distributed, they can be used to investigate the impact of twenty-five years of MNIST experiments on the reported testing performances. Our limited results unambiguously confirm the trends observed by Recht et al. [2018, 2019]: although the misclassification rates are slightly off, classifier ordering and model selection remain broadly reliable. We attribute this phenomenon to the pairing benefits of comparing classifiers on the same digits.

## 1 Introduction

The MNIST dataset [LeCun et al., 1994, Bottou et al., 1994] has been used as a standard machine learning benchmark for more than twenty years. During the last decade, many researchers have expressed the opinion that this dataset has been overused. In particular, the small size of its test set, merely 10,000 samples, has been a cause of concern. Hundreds of publications report increasingly good performance on this same test set. Did they overfit the test set? Can we trust any new conclusion drawn on this dataset? How quickly do machine learning datasets become useless?

The first partitions of the large NIST handwritten character collection [Grother and Hanaoka, 1995] had been released one year earlier, with a training set written by 2000 Census Bureau employees and a substantially more challenging test set written by 500 high school students. One of the objectives of LeCun, Cortes, and Burges was to create a dataset with similarly distributed training and test sets. The process they describe produces two sets of 60,000 samples. The test set was then downsampled to only 10,000 samples, possibly because manipulating such a dataset with the computers of the times could be annoyingly slow. The remaining 50,000 test samples have since been lost.

The initial purpose of this work was to recreate the MNIST preprocessing algorithms in order to trace back each MNIST digit to its original writer in NIST. This reconstruction was first based on the available information and then considerably improved by iterative refinements. Section 2 describes this process and measures how closely our reconstructed samples match the official MNIST samples. The reconstructed training set contains 60,000 images matching each of the MNIST training images. Similarly, the first 10,000 images of the reconstructed test set match each of the MNIST test set images. The next 50,000 images are a reconstruction of the 50,000 lost MNIST test images.[1]

> The original NIST test contains 58,527 digit images written by 500 different writers. In contrast to the training set, where blocks of data from each writer appeared in sequence, the data in the NIST test set is scrambled. Writer identities for the test set is available and we used this information to unscramble the writers. We then split this NIST test set in two: characters written by the first 250 writers went into our new training set. The remaining 250 writers were placed in our test set. Thus we had two sets with nearly 30,000 examples each.
>
> The new training set was completed with enough samples from the old NIST training set, starting at pattern #0, to make a full set of 60,000 training patterns. Similarly, the new test set was completed with old training examples starting at pattern #35,000 to make a full set with 60,000 test patterns. All the images were size normalized to fit in a 20 x 20 pixel box, and were then centered to fit in a 28 x 28 image using center of gravity. Grayscale pixel values were used to reduce the effects of aliasing. These are the training and test sets used in the benchmarks described in this paper. In this paper, we will call them the MNIST data.

Figure 1: The two paragraphs of Bottou et al. [1994] describing the MNIST preprocessing. The hsf4 partition of the NIST dataset, that is, the original test set, contains in fact 58,646 digits.

In the same spirit as [Recht et al., 2018, 2019], the rediscovery of the 50,000 lost MNIST test digits provides an opportunity to quantify the degradation of the official MNIST test set over a quarter-century of experimental research. Section 3 compares and discusses the performances of well known algorithms measured on the original MNIST test samples, on their reconstructions, and on the reconstructions of the 50,000 lost test samples. Our results provide a well controlled confirmation of the trends identified by Recht et al. [2018, 2019] on a different dataset.

## 2 Recreating MNIST

Recreating the algorithms that were used to construct the MNIST dataset is a challenging task. Figure 1 shows the two paragraphs that describe this process in [Bottou et al., 1994]. Although this was the first paper mentioning MNIST, the creation of the dataset predates this benchmarking effort by several months.[2] Curiously, this description incorrectly reports that the number of digits in the hsf4 partition, that is, the original NIST testing set, as 58,527 instead of 58,646.[3]

These two paragraphs give a relatively precise recipe for selecting the 60,000 digits that compose the MNIST training set. Alas, applying this recipe produces a set that contains one more zero and one less eight than the actual MNIST training set. Although they do not match, these class distributions are too close to make it plausible that 119 digits were really missing from the hsf4 partition.

The description of the image processing steps is much less precise. How are the 128x128 binary NIST images cropped? Which heuristics, if any, are used to disregard noisy pixels that do not belong to the digits themselves? How are rectangular crops centered in a square image? How are these square images resampled to 20x20 gray level images? How are the coordinates of the center of gravity rounded for the final centering step?

### 2.1 An iterative process

Our initial reconstruction algorithms were informed by the existing description and, crucially, by our knowledge of a mysterious resampling algorithm found in ancient parts of the Lush codebase: instead of using a bilinear or bicubic interpolation, this code computes the exact overlap of the input and output image pixels.[4]

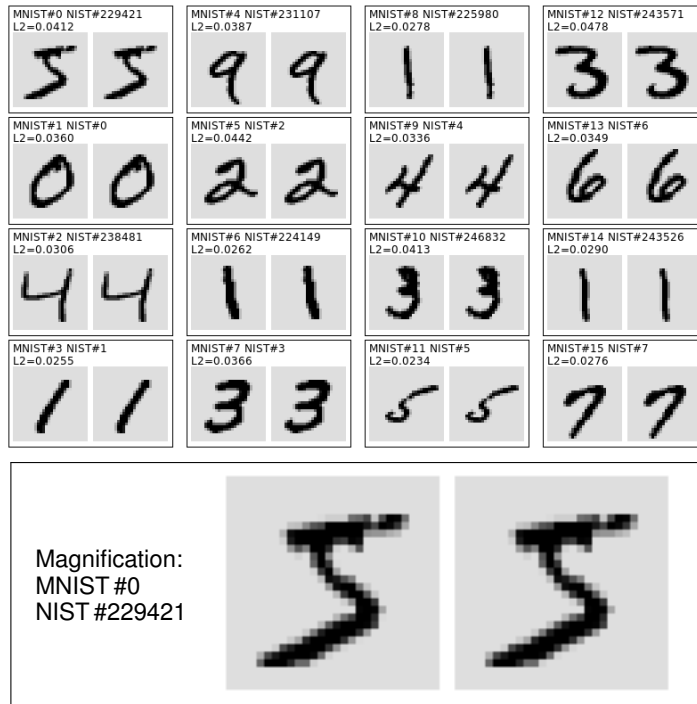

Figure 2: Side-by-side display of the first sixteen digits in the MNIST and QMNIST training set. The magnified view of the first one illustrates the correct reconstruction of the antialiased pixels.

Although our first reconstructed dataset, dubbed QMNISTv1, behaves very much like MNIST in machine learning experiments, its digit images could not be reliably matched to the actual MNIST digits. In fact, because many digits have similar shapes, we must rely on subtler details such as the anti-aliasing pixel patterns. It was however possible to identify a few matches. For instance we found that the lightest zero in the QMNIST training set matches the lightest zero in the MNIST training set. We were able to reproduce their antialiasing patterns by fine-tuning the initial centering and resampling algorithms, leading to QMNISTv2.

We then found that the smallest $L_2$ distance between MNIST digits and jittered QMNIST digits was a reliable match indicator. Running the Hungarian assignment algorithm on the two training sets gave good matches for most digits. A careful inspection of the worst matches allowed us to further tune the cropping algorithms, and to discover, for instance, that the extra zero in the reconstructed training set was in fact a duplicate digit that the MNIST creators had identified and removed. The ability to obtain reliable matches allowed us to iterate much faster and explore more aspects the image processing algorithm space, leading to QMNISTv3, v4, and v5. Note that all this tuning was achieved by matching training set images only.

This seemingly pointless quest for an exact reconstruction was surprisingly addictive. Supposedly urgent tasks could be indefinitely delayed with this important procrastination pretext. Since all good things must come to an end, we eventually had to freeze one of these datasets and call it QMNIST.

## 2.2 Evaluating the reconstruction quality

Although the QMNIST reconstructions are closer to the MNIST images than we had envisioned, they remain imperfect.

Table 2 indicates that about $0.25\%$ of the QMNIST training set images are shifted by one pixel relative to their MNIST counterpart. This occurs when the center of gravity computed during the last centering step (see Figure 1) is very close to a pixel boundary. Because the image reconstruction is imperfect, the reconstructed center of gravity sometimes lands on the other side of the pixel boundary, and the alignment code shifts the image by a whole pixel.

Table 1: Quartiles of the jittered distances between matching MNIST and QMNIST training digit images with pixels in range $0 \ldots 255$. A $L_2$ distance of 255 would indicate a one pixel difference. The $L_\infty$ distance represents the largest absolute difference between image pixels.

|  | Min | 25% | Med | 75% | Max |
|---|---|---|---|---|---|
| Jittered $L_2$ distance | 0 | 7.1 | 8.7 | 10.5 | 17.3 |
| Jittered $L_\infty$ distance | 0 | 1 | 1 | 1 | 3 |

Table 2: Count of training samples for which the MNIST and QMNIST images align best without translation or with a $\pm 1$ pixel translation.

| Jitter | 0 pixels | $\pm 1$ pixels |
|---|---|---|
| Number of matches | 59853 | 147 |

Table 3: Misclassification rates of a Lenet5 convolutional network trained on both the MNIST and QMNIST training sets and tested on the MNIST test set, on the 10K QMNIST testing examples matching the MNIST testing set, and on the 50k remaining QMNIST testing examples.

| Test on | MNIST | QMNIST10K | QMNIST50K |
|---|---|---|---|
| Train on MNIST | 0.82% ($\pm 0.2\%$) | 0.81% ($\pm 0.2\%$) | 1.08% ($\pm 0.1\%$) |
| Train on QMNIST | 0.81% ($\pm 0.2\%$) | 0.80% ($\pm 0.2\%$) | 1.08% ($\pm 0.1\%$) |

Table 1 gives the quartiles of the $L_2$ distance and $L_\infty$ distances between the MNIST and QMNIST images, after accounting for these occasional single pixel shifts. An $L_2$ distance of 255 would indicate a full pixel of difference. The $L_\infty$ distance represents the largest difference between image pixels, expressed as integers in range $0 \ldots 255$.

In order to further verify the reconstruction quality, we trained a variant of the Lenet5 network described by Le Cun et al. [1998]. Its original implementation is still available as a demonstration in the Lush codebase. Lush [Bottou and LeCun, 2001] descends from the SN neural network software [Bottou and Le Cun, 1988] and from its AT&T Bell Laboratories variants developed in the nineties. This particular variant of Lenet5 omits the final Euclidean layer described in [Le Cun et al., 1998] without incurring a performance penalty. Following the pattern set by the original implementation, the training protocol consists of three sets of 10 epochs with global stepsizes $10^{-4}$, $10^{-5}$, and $10^{-6}$. Each set starts with estimating the diagonal of the Hessian. Per-weight stepsizes are then computed by dividing the global stepsize by the estimated curvature plus 0.02. Table 3 reports insignificant differences when one trains with the MNIST or QMNIST training set or test with MNIST test set or the matching part of the QMNIST test set. On the other hand, we observe a more substantial difference when testing on the remaining part of the QMNIST test set, that is, the reconstructions of the lost MNIST test digits. Such discrepancies will be discussed more precisely in Section 3.

## 2.3 MNIST trivia

The reconstruction effort allowed us to uncover a lot of previously unreported facts about MNIST.

1. There are exactly three duplicate digits in the entire NIST handwritten character collection. Only one of them falls in the segments used to generate MNIST but was removed by the MNIST authors.

2. The first 5001 images of the MNIST test set seem randomly picked from those written by writers #2350-#2599, all high school students. The next 4999 images are the consecutive NIST images #35,000-#39,998, in this order, written by only 48 Census Bureau employees, writers #326-#373, as shown in Figure 5. Although this small number could make us fear for statistical significance, these comparatively very clean images contribute little to the total test error.

3. Even-numbered images among the 58,100 first MNIST training set samples exactly match the digits written by writers #2100-#2349, all high school students, in random order. The remaining images are the NIST images #0 to #30949 in that order. The beginning of this sequence is visible in Figure 2. Therefore, half of the images found in a typical minibatch of consecutive MNIST training images are likely to have been written by the same writer. We can only recommend shuffling the training set before assembling the minibatches.

4. There is a rounding error in the final centering of the 28x28 MNIST images. The average center of mass of a MNIST digits is in fact located half a pixel away from the geometrical center of the image. This is important because training on correctly centered images yields substantially worse performance on the standard MNIST testing set.

5. A slight defect in the MNIST resampling code generates low amplitude periodic patterns in the dark areas of thick characters. These patterns, illustrated in Figure 3, can be traced to a 0.99 fudge factor that is still visible in the Lush legacy code.[5] Since the period of these patterns depend on the sizes of the input images passed to the resampling code, we were able to determine that the small NIST images were not upsampled by directly calling the resampling code, but by first doubling their resolution, then downsampling to size 20x20.

6. Converting the continuous-valued pixels of the subsampled images into integer-valued pixels is delicate. Our code linearly maps the range observed in each image to the interval [0.0,255.0], rounding to the closest integer. Comparing the pixel histograms (see Figure 4) reveals that MNIST has substantially more pixels with value 128 and less pixels with value 255. We could not think of a plausibly simple algorithm compatible with this observation.

## 3 Generalization Experiments

This section takes advantage of the reconstruction of the lost 50,000 testing samples to revisit some MNIST performance results reported during the last twenty-five years. Recht et al. [2018, 2019] perform a similar study on the CIFAR10 and ImageNet datasets and identify very interesting trends. However they also explain that they cannot fully ascertain how closely the distribution of the reconstructed dataset matches the distribution of the original dataset, raising the possibility of the reconstructed dataset being substantially harder than the original. Because the published MNIST test set was subsampled from a larger set, we have a much tighter control of the data distribution and can confidently confirm their findings.

Because the MNIST testing error rates are usually low, we start with a careful discussion of the computation of confidence intervals and of the statistical significance of error comparisons in the context of repeated experiments. We then report on MNIST results for several methods: k-nearest neighbors (KNN), support vector machines (SVM), multilayer perceptrons (MLP), and several flavors of convolutional networks (CNN).

### 3.1 About confidence intervals

Since we want to know whether the actual performance of a learning system differs from the performance estimated using an overused testing set with run-of-the-mill confidence intervals, all confidence intervals reported in this work were obtained using the classic Wald method: when we observe $n_1$ misclassifications out of $n$ independent samples, the error rate $\nu = n_1/n$ is reported with confidence $1-\eta$ as

$$\nu \pm z\sqrt{\frac{\nu(1-\nu)}{n}}\,, \tag{1}$$

where $z = \sqrt{2}\operatorname{erfc}^{-1}(\eta)$ is approximately equal to 2 for a 95% confidence interval. For instance, an error rate close to $1.0\%$ measured on the usual 10,000 test example is reported as a $1\% \pm 0.2\%$ error rate, that is, $100 \pm 20$ misclassifications. This approach is widely used despite the fact that it only holds for a single use of the testing set and that it relies on an imperfect central limit approximation.

The simplest way to account for repeated uses of the testing set is the Bonferroni correction [Bonferroni, 1936], that is, dividing $\eta$ by the number $K$ of potential experiments, *simultaneously defined*

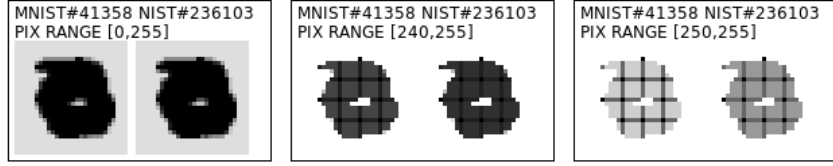

Figure 3: We have reproduced a defect of the original resampling code that creates low amplitude periodic patterns in the dark areas of thick characters.

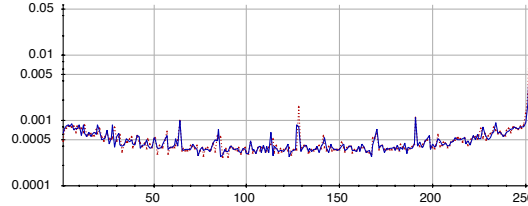

Figure 4: Histogram of pixel values in range 1-255 in the MNIST (red dots) and QMNIST (blue line) training set. Logarithmic scale.

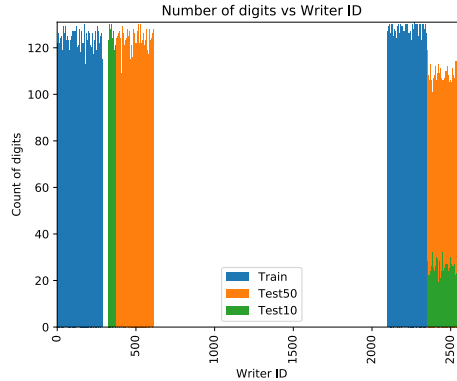

Figure 5: Histogram of Writer IDs and Number of digits written by the writer in MNIST Train, MNIST Test 10K and QMNIST Test 50K sets.

*before performing any measurement.* Although relaxing this simultaneity constraint progressively requires all the apparatus of statistical learning theory [Vapnik, 1982, §6.3], the correction still takes the form of a divisor $K$ applied to confidence level $\eta$. Because of the asymptotic properties of the erfc function, the width of the actual confidence intervals essentially grows like $\log(K)$.

In order to complete this picture, one also needs to take into account the *benefits* of using the same testing set. Ordinary confidence intervals are overly pessimistic when we merely want to know whether a first classifier with error rate $\nu_1 = n_1/n$ is worse than a second classifier with error rate $\nu_2 = n_2/n$. Because these error rates are measured on the same test samples, we can instead rely on a pairing argument: the first classifier can be considered worse with confidence $1-\eta$ when

$$\nu_1 - \nu_2 \; = \; \frac{n_{12} - n_{21}}{n} \; \geq \; z\frac{\sqrt{n_{12} + n_{21}}}{n} \, , \tag{2}$$

where $n_{12}$ represents the count of examples misclassified by the first classifier but not the second classifier, $n_{21}$ is the converse, and $z = \sqrt{2}\,\mathrm{erfc}^{-1}(2\eta)$ is approximately 1.7 for a 95% confidence. For instance, four additional misclassifications out of 10,000 examples is sufficient to make such a determination. This correspond to a difference in error rate of $0.04\%$, roughly ten times smaller than what would be needed to observe disjoint error bars (1). This advantage becomes very significant when combined with a Bonferroni-style correction: $K$ pairwise comparisons remain simultaneously

valid with confidence $1-\eta$ if all comparisons satisfy

$$n_{12} - n_{21} \quad \geq \quad \sqrt{2}\ \text{erfc}^{-1}\left(\frac{2\eta}{K}\right)\ \sqrt{n_{12} + n_{21}}$$

For instance, in the realistic situation

$$n = 10000\,,\ \ n_1 = 200\,,\ \ n_{12} = 40\,,\ \ n_{21} = 10\,,\ \ n_2 = n_1 - n_{12} + n_{21} = 170\,,$$

the conclusion that classifier 1 is worse than classifier 2 remains valid with confidence 95% as long as it is part of a series of $K \leq 4545$ pairwise comparisons. In contrast, after merely $K{=}50$ experiments, the 95% confidence interval for the absolute error rate of classifier 1 is already $2\% \pm 0.5\%$, too large to distinguish it from the error rate of classifier 2. We should therefore expect that repeated model selection on the same test set leads to decisions that remain valid far longer than the corresponding absolute error rates.[6]

## 3.2 Results

We report results using two training sets, namely the MNIST training set and the QMNIST reconstructions of the MNIST training digits, and three testing sets, namely the official MNIST testing set with 10,000 samples (MNIST), the reconstruction of the official MNIST testing digits (QMNIST10K), and the reconstruction of the lost 50,000 testing samples (QMNIST50K). We use the names TMTM, TMTQ10, TMTQ50 to identify results measured on these three testing sets after training on the MNIST training set. Similarly we use the names TQTM, TQTQ10, and TQTQ50, for results obtained after training on the QMNIST training set and testing on the three test sets. None of these results involves data augmentation or preprocessing steps such as deskewing, noise removal, blurring, jittering, elastic deformations, etc.

Figure 6 (left plot) reports the testing error rates obtained with KNN for various values of the parameter $k$ using the MNIST training set as reference points. The QMNIST50K results are slightly worse but within the confidence intervals. The best $k$ determined on MNIST is also the best $k$ for QMNIST50K. Figure 6 (right plot) reports similar results and conclusions when using the QMNIST training set as a reference point.

Figure 7 reports testing error rates obtained with RBF kernel SVMs after training on the MNIST training set with various values of the hyperparameters $C$ and $g$. The QMNIST50 results are consistently higher but still fall within the confidence intervals except maybe for mis-regularized models. Again the hyperparameters achieving the best MNIST performance also achieve the best QMNIST50K performance.

Figure 8 (left plot) provides similar results for a single hidden layer multilayer network with various hidden layer sizes, averaged over five runs. The QMNIST50K results again appear consistently worse than the MNIST test set results. On the one hand, the best QMNIST50K performance is achieved for a network with 1100 hidden units whereas the best MNIST testing error is achieved by a network with 700 hidden units. On the other hand, all networks with 300 to 1100 hidden units perform very similarly on both MNIST and QMNIST50, as can be seen in the plot. A 95% confidence interval paired test on representative runs reveals no statistically significant differences between the MNIST test performances of these networks. Each point in figure 8 (right plot) gives the MNIST and QMNIST50K testing error rates of one MLP experiment. This plot includes experiments with several hidden layer sizes and also several minibatch sizes and learning rates. We were only able to replicate the reported 1.6% error rate Le Cun et al. [1998] using minibatches of five or less examples.

Finally, Figure 9 summarizes all the experiments reported above. It also includes several flavors of convolutional networks: the Lenet5 results were already presented in Table 3, the VGG-11 [Simonyan and Zisserman, 2014] and ResNet-18 [He et al., 2016] results are representative of the modern CNN architectures currently popular in computer vision. We also report results obtained using four models from the TF-KR MNIST challenge.[7] Model TFKR-a[8] is an ensemble two VGG- and one ResNet-like models trained with an augmented version of the MNIST training set. Models

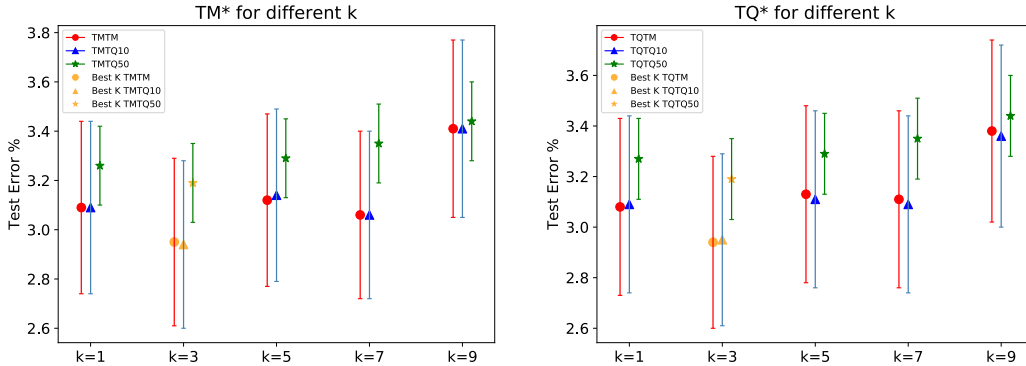

Figure 6: KNN error rates for various values of $k$ using either the MNIST (left plot) or QMNIST (right plot) training sets. Red circles: testing on MNIST. Blue triangles: testing on its QMNIST counterpart. Green stars: testing on the 50,000 new QMNIST testing examples.

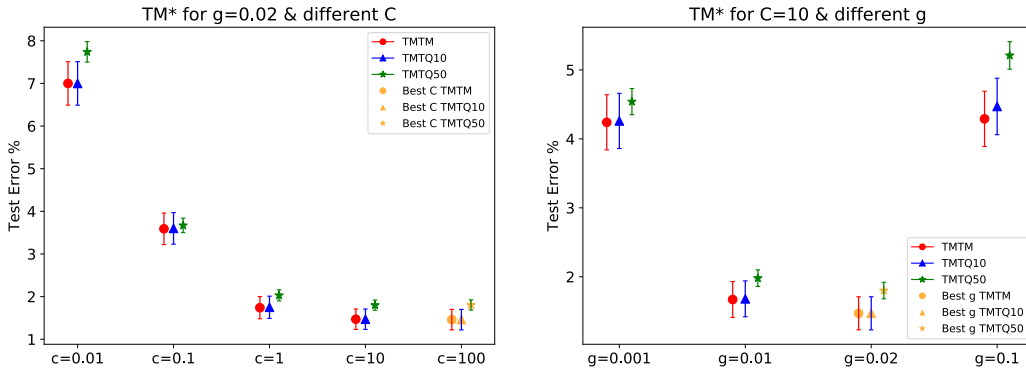

Figure 7: SVM error rates for various values of the regularization parameter $C$ (left plot) and the RBF kernel parameter $g$ (right plot) after training on the MNIST training set, using the same color and symbols as figure 6.

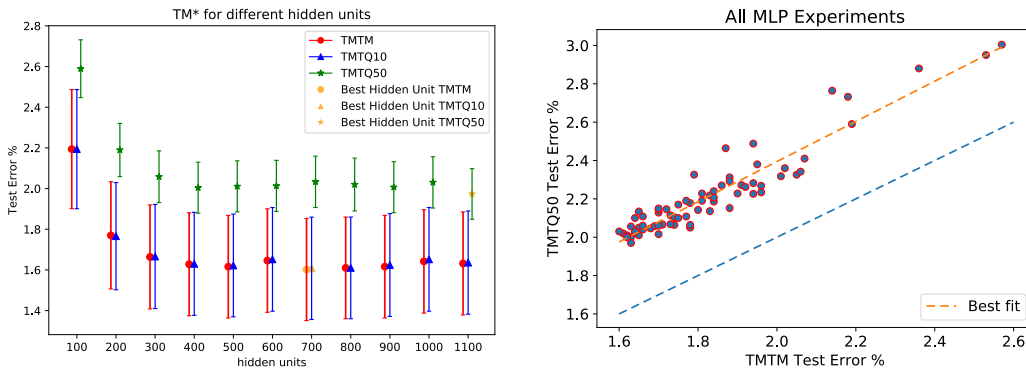

Figure 8: Left plot: MLP error rates for various hidden layer sizes after training on MNIST, using the same color and symbols as figure 6. Right plot: scatter plot comparing the MNIST and QMNIST50K testing errors for all our MLP experiments.

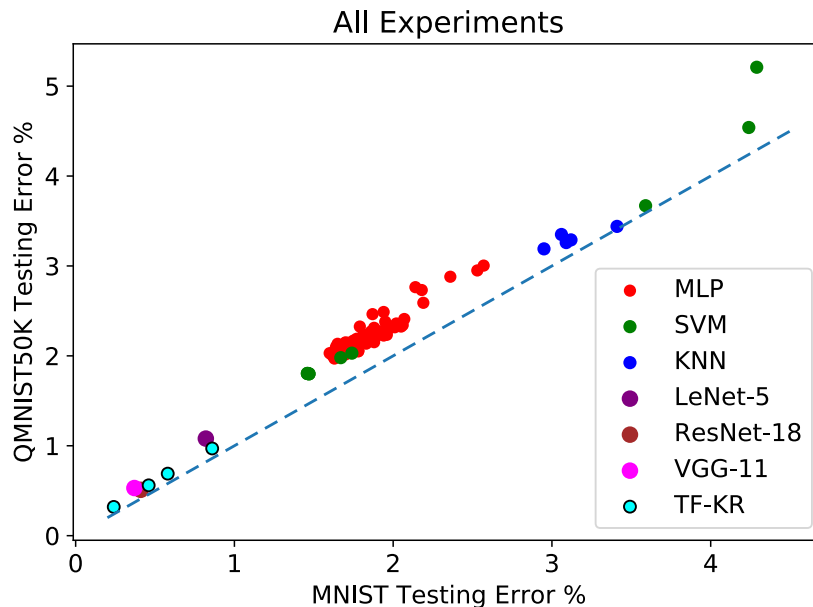

Figure 9: Scatter plot comparing the MNIST and QMNIST50K testing performance of all the models trained on MNIST during the course of this study.

TFKR-b[9], TFKR-c[10], and TFKR-d[11] are single CNN models with varied architectures. This scatter plot shows that the QMNIST50 error rates are consistently slightly higher than the MNIST testing errors. However, the plot also shows that comparing the MNIST testing set performances of various models provides a near perfect ranking of the corresponding QMNIST50K performances. In particular, the best performing model on MNIST, TFKR-a, remains the best performing model on QMNIST50K.

## 4   Conclusion

We have recreated a close approximation of the MNIST preprocessing chain. Not only did we track each MNIST digit to its NIST source image and associated metadata, but also recreated the original MNIST test set, including the 50,000 samples that were never distributed. These fresh testing samples allow us to investigate how the results reported on a standard testing set suffer from repeated experimentation. Our results confirm the trends observed by Recht et al. [2018, 2019], albeit on a different dataset and in a substantially more controlled setup. All these results essentially show that the "testing set rot" problem exists but is far less severe than feared. Although the repeated usage of the same testing set impacts absolute performance numbers, it also delivers pairing advantages that help model selection in the long run. In practice, this suggests that a shifting data distribution is far more dangerous than overusing an adequately distributed testing set.

## Acknowledgments

We thank Chris Burges, Corinna Cortes, and Yann LeCun for the precious information they were able to share with us about the birth of MNIST. We thank Larry Jackel for instigating the whole MNIST project and for commenting on this "cold case". We thank Maithra Raghu for pointing out how QMNIST could be used to corroborate the results of Recht et al. [2019]. We thank Ben Recht, Ludwig Schmidt and Roman Werpachowski for their constructive comments.

## Footnotes

[1]Code and data are available at https://github.com/facebookresearch/qmnist.

[2]When LB joined this effort during the summer 1994, the MNIST dataset was already ready.

[3]The same description also appears in [LeCun et al., 1994, Le Cun et al., 1998]. These more recent texts incorrectly use the names SD1 and SD3 to denote the original NIST test and training sets. And additional sentence explains that only a subset of 10,000 test images was used or made available, "*5000 from SD1 and 5000 from SD3.*"

[4]See https://tinyurl.com/y5z7qtcg.

[5]See https://tinyurl.com/y5z7abyt

[6]See [Feldman et al., 2019] for a different perspective on this issue.

[7]https://github.com/hwalsuklee/how-far-can-we-go-with-MNIST

[8]TFKR-a: https://github.com/khanrc/mnist

[9]TFKR-b: https://github.com/bart99/tensorflow/tree/master/mnist

[10]TFKR-c: https://github.com/chaeso/dnn-study

[11]TFKR-d: https://github.com/ByeongkiJeong/MostAccurableMNIST_keras

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
