[Supplementary Material]

# Supplementary Material

This section provides additional tables and plots.

Table 4: Misclassification rates of the best KNN model obtained when $k$ is set to 3. Model trained on both the MNIST and QMNIST training sets and tested on the MNIST test set, and the two QMNIST test sets of size 10,000 & 50,000 each.

| Test on | MNIST | QMNIST10K | QMNIST50K |
|---|---|---|---|
| Train on MNIST | 2.95% (±0.34%) | 2.94% (±0.34%) | 3.19% (±0.16%) |
| Train on QMNIST | 2.94% (±0.34%) | 2.95% (±0.34%) | 3.19% (±0.16%) |

Table 5: Misclassification rates of a SVM when hyperparameters $C$ = 10 & $g$ = 0.02. Training and testing schemes are similar to Table 4.

| Test on | MNIST | QMNIST10K | QMNIST50K |
|---|---|---|---|
| Train on MNIST | 1.47% (±0.24%) | 1.47% (±0.24%) | 1.8% (±0.12%) |
| Train on QMNIST | 1.47% (±0.24%) | 1.48% (±0.24%) | 1.8% (±0.12%) |

Table 6: Misclassification rates of an MLP with a 800 unit hidden layer. Training and testing schemes are similar to Table 4.

| Test on | MNIST | QMNIST10K | QMNIST50K |
|---|---|---|---|
| Train on MNIST | 1.61% (±0.25%) | 1.61% (±0.25%) | 2.02% (±0.13%) |
| Train on QMNIST | 1.63% (±0.25%) | 1.63% (±0.25%) | 2% (±0.13%) |

Table 7: Misclassification rates of a VGG-11 model. Training and testing schemes are similar to Table 4.

| Test on | MNIST | QMNIST10K | QMNIST50K |
|---|---|---|---|
| Train on MNIST | 0.37% (±0.12%) | 0.37% (±0.12%) | 0.53% (±0.06%) |
| Train on QMNIST | 0.39% (±0.12%) | 0.39% (±0.12%) | 0.53% (±0.06%) |

Table 8: Misclassification rates of a ResNet-18 model. Training and testing schemes are similar to Table 4.

| Test on | MNIST | QMNIST10K | QMNIST50K |
|---|---|---|---|
| Train on MNIST | 0.41% (±0.13%) | 0.42% (±0.13%) | 0.51% (±0.06%) |
| Train on QMNIST | 0.43% (±0.13%) | 0.43% (±0.13%) | 0.50% (±0.06%) |

Table 9: Misclassification rates of top TF-KR MNIST models trained on the MNIST training se and tested on the MNIST, QMNIST10K and QMNIST50K testing sets.

| Github Link | MNIST | QMNIST10K | QMNIST50K |
|---|---|---|---|
| TFKR-a | 0.24% (±0.10%) | 0.24% (±0.10%) | 0.32% (±0.05%) |
| TFKR-b | 0.86% (±0.18%) | 0.86% (±0.18%) | 0.97% (±0.09%) |
| TFKR-c | 0.46% (±0.14%) | 0.47% (±0.14%) | 0.56% (±0.07%) |
| TFKR-d | 0.58% (±0.15%) | 0.58% (±0.15%) | 0.69% (±0.07%) |

Figure 10: SVM error rates for various values of the regularization parameter $C$ (left plot) and the RBF kernel parameter $g$ (right plot) after training on the QMNIST training set. Red circles: testing on MNIST. Blue triangles: testing on its QMNIST counterpart. Green stars: testing on the 50,000 new QMNIST testing examples.

Figure 11: Left plot : MLP error rates for various hidden layer sizes after training on the QMNIST training set, using the same testing scheme as figure 10. Right plot: Paired test of MLPs with different hidden layer sizes and MLP with 700 hidden units (which performs best on MNIST test set). All of the MLPs used in this plot were trained and tested on MNIST.

Figure 12: Scatter plot comparing the best MNIST and QMNIST50K testing performance of all the classifiers trained on MNIST during the course of this study.