[Reviews · NeurIPS 2019]

Reviewer 1



The work represents a valuable contribution to the community's knowledge of its own history. The paper is well written and clear, rigorously reproducing the methods used to construct the most famous dataset in neural network research. I particularly enjoyed the narrative phrasing as lighthearted storytelling (e.g line 75).

Reviewer 2



The authors introduce a new version of the MNIST data set that they call QMNIST, which is the result of a very thoughtful and systematic analysis of existing materials used to build the original MNIST. I am impressed by the meticulous investigation carried out to recover the precise processing steps needed to generate MNIST examples from the original NIST images. The QMNIST data set is thus the product of a very accurate reconstruction of the original process (though the authors note that some minor discrepancies are still present). The authors then investigate whether the performance of popular classification methods measured using the new QMNIST test set actually differ from that measured on the original MNIST test set. Overall, I think this research is well conducted and presented in a very clear way. I consider it useful to try improving the testing methodology commonly used for MNIST benchmarking, since this data set has been often used as a reference for assessing the performance of new classification methods. I also agree with the authors that always using the same test set (as it has been done in the past 25 years for MNIST) might lead to biases in model evaluation, as also recently shown with other data sets (e.g., CIFAR-10). The advantage of MNIST is given by its simplicity, which allows for a more precise and controlled assessment compared to more complex data sets (such as the above-mentioned CIFAR-10, or even ImageNet). However, such simplicity is probably also the weakness of MNIST, since differences in test accuracy are often negligible, and can be due to idiosyncrasies encoded in particular test images. Moreover, given the recent trend in evaluating state-of-the-art models on more challenging, large-scale object recognition tasks, I wonder how relevant this would really be for informing and guiding future research. So, from a scientific point of view my feeling is that the significance of this study for the NeurIPS audience might be low. However, I also see some merit in the historical perspective taken by the authors and feel that this paper would be of interest for many people attending NeurIPS. I still wonder whether we should really care about the fine-grained details that the authors discuss in the text, but if the authors argue that they received positive feedback about it I would not insist on this point. I still also wonder whether providing a “definitive” test set might not be the best way to improve model evaluation: maybe a better approach should be to use a cross-validation procedure, where training, validation and test sets are randomly sampled. Maybe the authors could mention that the additional information they are providing can be indeed used to implement more controlled cross-validation schemes, for example by leaving out specific writers.

Reviewer 3



Originality: the paper contains no original technical work. The original work on MNIST is correctly cited, as is work to related papers by Recht. Quality: Excellent. I commend the authors in their very careful detective work in reconstructing MNIST. This kind of careful data provenance on a very common dataset should be rewarded. Clarity: Very good. I understood how the authors did their example matching, and the comparison experiments are clear. Significance: The results are moderately important. If you had asked me in 2010 whether we would live in a world where ImageNet has <3% top-5 error, while people still used MNIST as a benchmark, I would have laughed at you. However, that is the world we're in. Given the prevalence of the benchmark, it's important to know how much overfitting there is by the community.

[Author Response · NeurIPS 2019]

Dear Reviewers,

We would like to thank you for your careful assessment of our paper. In order to improve the paper we intend to make the following changes:

1. One of the goals of Section 3.1 in the paper is to determine whether *customary confidence intervals* become poor indicators when one overuses the testing set. This is why we prefer using the simple and popular Wald approach over more precise approaches such as binomial tails or empirical Bernstein bounds. We intend to make this clear. The second goal of section 3.1 is to show how pairing improves the confidence intervals so much that Bonferroni-style corrections have far less impact. We intend to re-word this argument to emphasize the assumptions we are making and discuss how the argument resists when the assumptions are weakened.

2. We also intend to add additional experimental results. In particular we would like to have a modern CNN (ReLU instead of sigmoids, many more feature maps) and a system that relies on manually crafted features. However it is clear that whatever list of systems we test will remain very small in comparison to the countless results published in the literature. This is why we are looking forward to seeing more results using the QMNIST testing set.

3. We believe that the 2.3 MNIST trivia section is important and needs to be documented. In fact we received substantial positive feedback about this section and inquiries for even more information.

Regards,

Authors

[Meta-Review · NeurIPS 2019]

Reviewers and the area chair agree that this paper will attract interest at the conference, and will be beneficial for future research. As this submission notes, Ben Recht and co-authors did similar work with other datasets, published recently. The citation for "Do ImageNet Classifiers Generalize to ImageNet?" should be updated: @InProceedings{pmlr-v97-recht19a, title = {Do {I}mage{N}et Classifiers Generalize to {I}mage{N}et?}, author = {Recht, Benjamin and Roelofs, Rebecca and Schmidt, Ludwig and Shankar, Vaishaal}, booktitle = {Proceedings of the 36th International Conference on Machine Learning}, pages = {5389--5400}, year = {2019}, editor = {Chaudhuri, Kamalika and Salakhutdinov, Ruslan}, volume = {97}, series = {Proceedings of Machine Learning Research}, address = {Long Beach, California, USA}, month = {09--15 Jun}, publisher = {PMLR}, pdf = {http://proceedings.mlr.press/v97/recht19a/recht19a.pdf}, url = {http://proceedings.mlr.press/v97/recht19a.html}, }